# OpenReview forum: "Improved Representation Steering for Language Models"
_NeurIPS.cc/2025/Conference — NeurIPS 2025 spotlight_

### Official Review · Reviewer_zyK7 · 2025-06-21

**Clarity:** 3
**Significance:** 3
**Originality:** 3
**Rating:** 5
**Confidence:** 4

**Summary:**

This paper studies representation steering of language models (LMs) where the goal is to introduce or suppress a given concept. The paper proposes a novel bidirectional preference-optimization objective called Reference-free Preference Steering (RePS). It extends SimPO and removes the constraint of staying close to the reference model. The objective is bidirectional in the sense that is the sum of two objectives: likelihood differences for positive steering and for negative steering. The proposed objective is used in conjunction with different intervention methods such as steering vectors, LoReFT, LoRA and evaluated on AXBENCH. It is compared to other baseline methods like language modeling and BiPO. It outperforms the other baselines but still performs worse than prompting.

**Questions:**

1 - How invasive are these interventions? Do they affect the other capabilities of the model? Does the model still perform similarly on common LM benchmarks?

2 - How do you select the ranges of hyperparameters that you search on? Do you use the same ranges for combinations of intervention methods/objective?

3 - How extensive was your prompt engineering for the attack experiment in section 5.4?

**Ethical Concerns:**

["NO or VERY MINOR ethics concerns only"]

**Final Justification:**

The authors have answered all the questions, provided clarifications and agreed to include the additional results in the final version. As my concerns were properly addressed, I have increased my score to 5.

**Limitations:**

yes

**Quality:**

4

**Strengths And Weaknesses:**

Strengths:
- The method is evaluated with various intervention methods and models of different sizes. Moreover, it is compared with other strong baselines.
- RePS outperforms the baselines besides prompting on AXBENCH almost across all model sizes and values of r.
- The paper is mostly self-contained in the sense that the extensive background and prior methods required to truly situate the contributions are all explained in the paper. This might help people not very familiar with the field appreciate the proposed method. Moreover the paper is structured and written in a clear manner without unnecessary verbosity. There are a few exceptions mentioned in the Weaknesses section below.

Weaknesses:
- The method introduces significant overhead but seems to still perform worse than prompting.
- The techniques used in section 5.4 for the attack experiments might not be representative of prompting. There are good prompting techniques tailored to jailbreaking that would make decent baselines but are not mentioned in that section.

minor comments:
It might be helpful to add error bars to the tables

---

> ### Author Rebuttal · Authors · 2025-07-29
>
> Thanks for your comments! We address them below. We further clarify the advantages of representation-based steering methods against prompting.
>
> > **Q1:** The method introduces significant overhead but seems to still perform worse than prompting.
>
> **A1:** We feel the trade-offs are nuanced but ultimately show that RePS has significant value:
>
> - **First, in the short‑context regime the memory and compute overhead are actually much lower for intervention‑based methods.** Given a fixed‑length steering prompt of length L, we need to store the per‑layer KV cache for the prompt and attend to L tokens when decoding each token. By contrast, steering vectors require storing only a single d‑dimensional vector and performing a simple vector‑addition step (`b*s*d` FLOPs where `b` is batch size, `s` is sequence length and `d` is model dimension). Considering absolute memory consumption and FLOPs, representation‑based steering therefore outperforms prompting. When the context length increases, the percentage‑wise overhead of the steering‑prompt prefix decreases dramatically simply because the dominating factor becomes the generated sequence or the long‑context prompt. Similarly, the FLOPs overhead from multi‑head attention constitutes less than 10 % of the model’s total FLOPs for generating a single token [1], so the gains are small. We will add these discussions to the Appendix in the next revision.
>
> - Secondly, we agree that although we significantly narrow the gap, representation-based steering still underperforms compared against prompting. **However, when we consider the case of feature suppression, we do see representation-based approaches are better when guardrailing against attacks** (i.e., attackers trying to elicit the features) when suppressing a features (as shown in Table 3). We additionally show that (in Appendix Q), prompting-base feature suppression is susceptible to leak out system prompts when used for defending attacks.
>
>
> > **Q2:** The techniques used in section 5.4 for the attack experiments might not be representative of prompting. There are good prompting techniques tailored to jailbreaking that would make decent baselines but are not mentioned in that section. How extensive was your prompt engineering for the attack experiment in section 5.4?
>
> **A2:** Thanks for pointing this out. Because of compute and time‑budget limitations, we experimented only with 4o‑mini‑generated attacks and many‑shot jailbreak attacks (up to 300 shots, producing an attack prompt at the context‑window limit of the models we tested). We agree these experiments are not exhaustive, but we believe they provide concrete evidence of the advantages of representation‑based steering (see Table 3) and highlight potential failure cases of prompting‑based guardrailing, which can accidentally leak the system prompt (see Appendix Q).
>
> > **Q3:** minor comments: It might be helpful to add error bars to the tables
>
> **A3:** Thanks for the suggestions! We will refine our result tables.
>
> > **Q4:** How invasive are these interventions? Do they affect the other capabilities of the model? Does the model still perform similarly on common LM benchmarks?
>
> **A4:** Great question! **This is precisely why we use AxBench and its evaluation paradigm to report our results.** AxBench measures three scores: (1) a *concept score* that indicates whether the intervention was applied successfully; (2) an *instruction score* that reflects whether the intervened model is still following the instruction; and (3) a *fluency score* that checks whether the LM is generating fluent responses. Each generation is then scored by taking the harmonic mean of these three scores, ensuring we reward interventions that are effective without compromising the model’s ability to follow instructions.
>
> We followed the AxBench evaluation settings and did not test on general LM benchmarks because the goal of interventions is not to enhance or maintain overall performance but to steer models when needed. Likewise, AxBench is not an instruction‑following benchmark for LLMs; it is designed to assess concept‑based detection and steering of various representation‑based methods, not to evaluate general instruction‑following ability. This distinction aligns with the focus of different steering methods.
>
> > **Q5:** How do you select the ranges of hyperparameters that you search on? Do you use the same ranges for combinations of intervention methods/objective?
>
> **A5:** Yes! We use the same range of hyperparameters for each combination of method and objective and ensure we allocate the same tuning budget to each of them. Details are provided in Appendix E. As we emphasize in our responses to other reviewers, this is not because it is hard to find optimal hyperparameters (see Fig. 16 for the performance‑variance plot) but rather to give each method a fair shot.
>
> [1] Mu et. al., 2023 Learning to Compress Prompts with Gist Tokens, https://arxiv.org/abs/2304.08467

---

> > ### Comment · Reviewer_zyK7 · 2025-08-02
> >
> > Thank you for your answer. The authors have addressed almost all the issues raised in the review and answered all the questions. Concerning the jailbreaking part, there are manual attacks that wouldn't necessarily require a lot of compute. Beyond AxBench, I would still recommend evaluating the general instruction following capabilities of the model to see how they are affected by the interventions. This could be of interest to the community. It doesn't have to be in the main text but could fit in the appendix.

---

> > > ### Author Response · Authors · 2025-08-03
> > > **Thank you for the response.**
> > >
> > > Thanks for your follow-up and suggestions regarding the next steps! We will consider including these additional experiments (i.e., manual attacks on suppression and capability measurement with interventions in place) in our next revision.

---

### Official Review · Reviewer_pxDh · 2025-06-27

**Clarity:** 4
**Significance:** 3
**Originality:** 2
**Rating:** 5
**Confidence:** 3

**Summary:**

The paper presents Reference-free Preference Steering (RePS) for controlling LLM outputs via internal representation manipulation. The core idea of RePS is that when training the intervention $\Phi$, utilizing SimPO inspired training objective as opposed to BiPO that uses reference model likelihood. The method is evaluated on AxBench with Gemma models, across 3 intervention types, steering vectors, LoRA and ReFT. The method outperforms existing methods in steering and supression and is effective for larger model steering.

**Questions:**

Q1. Clarify the underperformance of LoRA and ReFT: The results show that SV consistently outperforms LoRA and ReFT, especially on larger models (Table 1). Could the authors provide a deeper analysis of why these low-rank methods underperform? For example, is this due to limitations in the parameterization, training data, or hyperparameter tuning?

Q2. Can we quantify the computational cost and propose strategies for resource-constrained settings, such as fewer runs or heuristic-based tuning?

**Ethical Concerns:**

["NO or VERY MINOR ethics concerns only"]

**Final Justification:**

Overall this work proposes a simple approach for concept steering, and the authors clarified most questions in the original review. I will keep my original positive evaluation of the work, although with less confidence as I am not well-versed in the field of mechanistic interpretability.

**Limitations:**

yes

**Quality:**

3

**Strengths And Weaknesses:**

Overall, I think the work is a well executed, solid research paper (reserving the fact that I am not familiar with representation steering literature).

- The idea of RePS is simple and thus broadly applicable. It’s motivation is well grounded on existing discussions on preference optimization, thus can be intuitively understood. Yet its empirical performance seems to be substantially better than comparable methods in both concept steering and suppression .
- Experimental design is robust, using AxBench across a range of model scales. Inclusion of multiple intervention types enhances reliability of results.
- Writing is well-structured, with clear exposition of preliminaries, methodology, and mathematical formulation of RePS and the intervention types used.
- The paper lacks a detailed discussion of the limitations of RePS, particularly regarding the underperformance of LoRA and ReFT compared to SV on larger models. This raises questions about the generalizability of RePS across different intervention types.
- The hyperparameter search (72 runs for smaller models, 168 for larger ones) suggests significant computational demands, which limits practical application of RePS for resource-constrained setting.

Minor Typo: in equation (3), in the second term, $y^l | x$ should be changed to $y | x$

---

> ### Author Rebuttal · Authors · 2025-07-29
>
> Thanks for your comments! We address them below. We further clarify the underperformance of LoRA and ReFT and our hyperparameter-tuning procedure.
>
> > **Q1:** Clarify the underperformance of LoRA and ReFT: The results show that SV consistently outperforms LoRA and ReFT, especially on larger models (Table 1). Could the authors provide a deeper analysis of why these low-rank methods underperform? For example, is this due to limitations in the parameterization, training data, or hyperparameter tuning?
>
> **A1:** Great question! We believe this is largely due to two factors that warrant further exploration.
> - **Negative‑intervention design.** Our design is driven by interpretability work on rank‑1 subspace edits (Eqn 7). Both LoRA and LoReFT exhibit non‑linear behavior when removing features in high‑rank settings, leaving the definition of negative steering somewhat unjustified. Following prior work, we simply apply negative coefficients to the learned LoRA and LoReFT weights, but whether this is optimal remains under‑explored. Existing studies have also shown that both methods are highly sensitive to the steering factors [1].
>
> - **Limited training data in AxBench.** AxBench focuses on concept steering with fewer than 100 training pairs, so high‑rank interventions are more prone to overfitting. Future work could compare these methods in domains with more complex steering targets or larger datasets—for example, personalization datasets such as LAMP [2].
>
>
> > **Q2:** The hyperparameter search (72 runs for smaller models, 168 for larger ones) suggests significant computational demands, which limits practical application of RePS for resource-constrained setting.
>
> **A2:** Thanks for raising this issue! **We want to emphasize that this is actually a feature of our paper, not a bug of RePS**, as discussed extensively in Appendix D. The extensive hyperparameter tuning across all conditions—using the same grid or the same budget for each pair of intervention type and training objective—ensures that we study the true generalization of our training objective.
>
> **In fact, we encourage researchers in this field to adopt our comprehensive hyperparameter‑tuning procedures** to give all methods a fair chance. With only a small grid search, one method could simply be lucky enough to outperform another, especially since steering vectors are usually trained in a low‑data regime.
>
> Moreover, RePS is relatively stable across our hyperparameter runs; we provide a detailed performance‑variance analysis in Appendix D (e.g., see Fig. 16). This means it is not particularly costly to select a good hyperparameter for a new dataset.
>
> > **Q3:** Minor Typo: in equation (3)
>
> **A3:** Thanks! We will correct the typo in our next revision.
>
> > **Q4:** Can we quantify the computational cost and propose strategies for resource-constrained settings, such as fewer runs or heuristic-based tuning?
>
> **A4:** Yes! Again, most of our compute budget was spent on an extensive hyperparameter grid search for all methods (168 runs for each condition to select the best setting), as reported in Appendix E. This is not because it is hard to find optimal hyperparameters (see Fig. 16 for the performance‑variance plot) but to ensure we give each method a fair chance.
> We will include our computational costs in the next revision. All of our training runs can be carried out on a single 80 GB A100 GPU. To give a rough estimate, it **takes less than one minute to train** a single steering vector for a concept, and it **costs less than $0.01** to create the preference‑pair training data for that vector. We will provide a detailed analysis in the next revision.
>
> [1] Zhang et. al., 2023 Composing Parameter-Efficient Modules with Arithmetic Operations https://arxiv.org/abs/2306.14870
>
> [2] Salemi et. al., 2023, LaMP: When Large Language Models Meet Personalization, https://arxiv.org/abs/2304.11406

---

> > ### Comment · Reviewer_pxDh · 2025-08-03
> >
> > Thank you for the response. This clarifies most of the questions in the original review. I will maintain my positive evaluation of the work.

---

### Official Review · Reviewer_7WLo · 2025-07-02

**Clarity:** 3
**Significance:** 2
**Originality:** 2
**Rating:** 5
**Confidence:** 3

**Summary:**

The proposed method, Reference-free Preference Steering (RePS), introduces a new bidirectional preference optimization objective to improve representation steering in language models. RePS simultaneously optimizes for two contrasting behaviors: concept steering and concept suppression. For concept steering, when interventions are applied positively, RePS up-weights the reward of steered behavior by increasing the likelihood of the desired steered response (winning response) compared to the original response (losing response). For suppression, when interventions are applied negatively, it learns to null out any projection along the steering direction from the representation. The authors implement and train three parameterizations of RePS and evaluate them using AXBENCH, a benchmark for model steering. Across Gemma models ranging from 2B to 27B parameters, RePS outperforms other steering methods trained with traditional language modeling objectives and closes the gap with prompt-based steering. RePS also demonstrated more robustness to prompt-based jailbreaking attacks compared to prompt-only defenses.

**Questions:**

The paper presents concept suppression results only for rank-1 steering vectors trained with RePS and the language modeling objective, omitting evaluations for other intervention types such as LoRA and LoReFT, as well as alternative objectives like BiPO (e.g., in Table 2). Could the authors clarify why these results were not included?

**Ethical Concerns:**

["NO or VERY MINOR ethics concerns only"]

**Final Justification:**

The authors have addressed my concerns and provided valuable additional experimental results. Therefore, I raise my score to 5.

**Limitations:**

Yes, the limitations are included in Section 6.

**Quality:**

3

**Strengths And Weaknesses:**

**Strengths**

1. The paper is clearly written and easy to follow.
2. The proposed method, RePS, is a new reference-free, bidirectional preference‑optimization objective that jointly does concept steering and suppression.
3. RePS incorporates a factor‑sampling trick during training to stabilize optimization.
4. The method outperforms existing steering methods across various Gemma model sizes.

**Weaknesses**

1. The experimental evaluation of concept suppression is somewhat limited. In particular, the paper does not report suppression results for other intervention-based methods such as LoRA and LoReFT, nor does it include comparisons using the BiPO objective in Table 2. Including these results would offer valuable insight into whether the dual capabilities of RePS, both steering and suppression, generalize across different intervention types and training objectives.
2. It is mentioned in lines 234 and 235, that finding the optimal parameters (including the optimal steering layers) required extensive grid searches (up to 168 runs per objective–method for gemma-3 models). This limits the practical adoption and scalability of the proposed approach.

---

> ### Author Rebuttal · Authors · 2025-07-29
>
> Thanks for your comments! We address them below. We provide additional results on LoRA and LoReFT and discuss our hyperparameter-tuning procedure in detail.
> > **Q1**:  The experimental evaluation of concept suppression is somewhat limited. The paper presents concept suppression results only for rank-1 steering vectors trained with RePS and the language modeling objective, omitting evaluations for other intervention types such as LoRA and LoReFT, as well as alternative objectives like BiPO (e.g., in Table 2). Could the authors clarify why these results were not included?
>
> **A1:** Great suggestions! These are mainly two reasons why we did not prioritize these results:
> - **First, steering performance is our primary focus.** We excluded BiPO from further studies because its positive‑steering performance lags significantly behind (Table 1), making it impractical for other steering applications such as suppression.
> - **Second, the negative‑steering intervention in Eqn (7) is designed only for rank‑1 steering**; its objective is far better motivated in that setting. For LoRA and LoReFT, negative steering during training remains largely under‑explored—previous work typically concentrates on zero‑shot adaptation of negative weights to test whether interventions can be composed algorithmically.
> To provide additional insight, we trained LoRA and LoReFT on 2B and 9B models, applied the interventions negatively using the training‑time steering factors (i.e., multiplying each rank by the same negative factor), and evaluated them in the same way as the rank‑1 steering reported in Table 2:
>
> | Method | Condition | 2B L20 | 2B L31 | 9B L20 | 9B L31 |
> | ------ | --------- | -----: | -----: | -----: | -----: |
> | LoRA   | Lang      |  0.184 |  0.117 |  1.108 |  1.108 |
> |        | RePS      |  0.000 |  0.001 |  0.822 |  0.808 |
> | LoReFT | Lang      |  0.522 |  0.712 |  0.603 |  0.704 |
> |        | RePS      |  0.697 |  0.727 |  0.688 |  0.687 |
>
>
> **Our results suggest that RePS benefits LoReFT, whereas the LoRA outcomes are somewhat mixed.** In particular, RePS interventions failed on the 2B LoRA models when applied negatively. We believe this is largely because LoRA is highly sensitive to inference‑time coefficient adjustments, as shown in previous work [1]—LoRA simply breaks when large negative factors are applied. Consequently, we recommend that future work design better negative interventions for high‑rank methods. We will include these additional results and discussions in the Appendix.
>
> > **Q2:** It is mentioned in lines 234 and 235, that finding the optimal parameters (including the optimal steering layers) required extensive grid searches (up to 168 runs per objective–method for gemma-3 models). This limits the practical adoption and scalability of the proposed approach.
>
> **A2:** Thanks for raising this issue! **We want to emphasize that this is actually a feature of our paper, not a bug of RePS**, as discussed extensively in Appendix D. The extensive hyperparameter tuning across all conditions—using the same grid or the same budget for each pair of intervention type and training objective—ensures that we study the true generalization of our training objective.
>
> **In fact, we encourage researchers in this field to adopt our comprehensive hyperparameter‑tuning procedures** to give all methods a fair chance. With only a small grid search, one method could simply be lucky enough to outperform another, especially since steering vectors are usually trained in a low‑data regime.
>
> Moreover, RePS is relatively stable across our hyperparameter runs; we provide a detailed performance‑variance analysis in Appendix D (e.g., see Fig. 16). This means it is not particularly costly to select a good hyperparameter for a new dataset.
>
> [1] Zhang et. al., 2023 Composing Parameter-Efficient Modules with Arithmetic Operations https://arxiv.org/abs/2306.14870

---

> > ### Comment · Reviewer_7WLo · 2025-08-03
> >
> > I appreciate the authors’ detailed response and the additional experimental results provided. The authors have addressed almost all of my concerns. I strongly encourage including these new results and discussions in the camera-ready version.

---

### Official Review · Reviewer_E2Rg · 2025-07-03

**Clarity:** 3
**Significance:** 3
**Originality:** 3
**Rating:** 4
**Confidence:** 1

**Summary:**

The most straightforward way to steer language model is through prompting to provide fine-grained and interpretable control over model generations by variously changing model inputs, weights, or representations to adjust behavior. Another possibility is through a indirect way by adjusting weights or representations, which is often less effective. In this work, Reference-free Preference Steering (RePS), a bidirectional preference-optimization objective is proposed for jointly concept steering and suppression. Evaluated with AXBENCH, the model showed significant improvement among representation based steering and narrowed the gap with prompting.

**Questions:**

- As a researcher outside "Mechanistic Interpretability" area, my main question is about the motivation to study the implicit steering approach given that the LLM is a natural language interface. Given that the most natural way is prompting, it seems we just need to keep improving the instruction following capability of the model. And if we got a "omni" model that can follow instruction perfectly, it seems the LLM steering problem is solved. It would be good to provide more context and motivations from audience outside this area, for example, what's the pros and cons when comparing with prompting.

**Ethical Concerns:**

["NO or VERY MINOR ethics concerns only"]

**Final Justification:**

The previous concerns is mostly about the motivation of Mechanistic Interpretability area itself. Now that the concern has been resolved and I would like to raise the rating

**Limitations:**

Yes

**Quality:**

3

**Strengths And Weaknesses:**

### Strength

- The proposed method is clearly presented in the paper and easy to understand.
- There is significant improvement within the family of representation based steering category, demonstrating the effectiveness of this approach.

### Weakness
- The self-contain perspective can be improved. As a researcher outside "Mechanistic Interpretability" area, it's a bit challenging to understand the context and motivation about why representation based steering.

---

> ### Author Rebuttal · Authors · 2025-07-29
>
> Thanks for your comments! We address them below. We further clarify our motivations behind improving representation-based steering methods.
> > **Q1**: The self-contain perspective can be improved. As a researcher outside "Mechanistic Interpretability" area, it's a bit challenging to understand the context and motivation about why representation based steering.
> **A1:** Great suggestions! We will improve our abstract and introduction and clarify our motivations early. Here is a quick summary about our motivations.
> - **Interpretability and flexibility**. The goal of representation steering—particularly rank‑1 steering vectors—is to provide more interpretable and flexible inference‑time LM control, where prompting falls short. For example, you can flexibly inject steering vectors at inference time, whereas prompting would require re‑prompting the model, incurring significant KV‑memory overhead. Given its interpretable nature, one could also build on it to explore tasks such as feature suppression (as we show in Tables 2 and 3) or composed steering.
> - **RePS closes the performance gap and makes representation-based steering competitive again**. Prior to RePS, previous work showed that representation‑based steering methods lagged far behind, making them unattainable as alternative approaches. In particular, the preference‑based approach BiPO fails almost catastrophically on AxBench, a large‑scale LM‑steering benchmark. RePS closes the gap significantly (achieving a 2.7× improvement over BiPO and outperforming language‑modeling training objectives). By narrowing the gap, we hope RePS will provide insights for future improvements in representation‑based steering.
> - **Representation-based steering is safer than prompting when guardrailing against attacks**. Representation‑based approaches shine under realistic jailbreaking scenarios, as shown in Table 3. In feature‑suppression under attacks (i.e., steering LMs to avoid mentioning certain concepts), representation‑based steering consistently outperforms prompting and survives jailbreak attempts. We additionally show (in Appendix Q) that prompt‑based feature suppression is susceptible to leaking system prompts when used for defense against attacks.
> We hope these clarifications help!
>
> > **Q2**:  If we got a "omni" model that can follow instruction perfectly, it seems the LLM steering problem is solved. It would be good to provide more context and motivations from audience outside this area, for example, what's the pros and cons when comparing with prompting.
>
> **A2:** Great questions! Unlike previous work, which primarily explores ways to improve steering performance, we systematically investigate cases where representation‑based steering can outperform prompting and offer unique benefits. Here are our main arguments for why representation‑based steering is not attempting to bet against the scaling laws of LM performance:
> - **Representation‑based guardrailing methods are safer.** System prompts always leak, which can expose dangerous information about model behaviors. By contrast, as we show in Sec. 5.3 and Appendices K, L, M, and N, representation‑based feature‑suppression methods are much more robust, especially against jailbreak attacks. Moreover, system prompts can be leaked, whereas representation‑based steering is much harder to reverse‑engineer.
> - **Personalized LMs.** Although personalized history can be jammed into the prompt as prefix KV caches, this quickly incurs substantial overhead due to large memory requirements. Offloading the personalized prompt into static, small steering vectors not only saves memory and compute—reducing attention calculations over KV caches to a simple rank‑1 vector addition—but can also be more precise because of their trainable nature. As we show in the paper, training such steering vectors works even in a low‑resource data regime.

---

> > ### Comment · Reviewer_E2Rg · 2025-08-02
> > **Thanks for providing the additional context**
> >
> > Thanks the authors for providing the additional context about the Mechanistic Interpretability. Overall it sounds like the goal is to compress the prompt instruction into compacted embedding space, and use these as contextual signals. This is an well-adopted idea in terms of recommendation system to compress user's engagement history into a embedding, e.g. [1]. It's good to know that different area shared the same idea. I adopted my rating accordingly.
> >
> > [1] Lyu, Wenhan, et al. "DV365: Extremely Long User History Modeling at Instagram." KDD (2025).

---

### Decision · Program_Chairs · 2025-09-17

**Decision:**

Accept (spotlight)

**Comment:**

The paper builds off of relatively recent works such as SimPO and BiPO to suggest a method RePS to do better controllable generation of LLMs. While many works in this area require (data or model) references or explicit learning of constraint, this particular method is appealing as it seems to address a basic property of the task: desired control behaviors are not necessarily likely under a learned reference. Empirically as well, the reviewers all agree that this paper is well executed and has potential to be integrated into a wide variety of contexts. There is no doubt that this work, given all the changes promised by the authors, should be accepted.